# Explicit and Intrinsic Intention to Receive COVID-19 Vaccination among Heterosexuals and Sexual Minorities in Taiwan

**DOI:** 10.3390/ijerph18147260

**Published:** 2021-07-07

**Authors:** Yen-Ju Lin, Yu-Ping Chang, Wen-Jiun Chou, Cheng-Fang Yen

**Affiliations:** 1Department of Psychiatry, School of Medicine, College of Medicine, Kaohsiung Medical University, Kaohsiung 80708, Taiwan; 1040457@kmuh.org.tw; 2Department of Psychiatry, Kaohsiung Medical University Hospital, Kaohsiung 80708, Taiwan; 3School of Nursing, The State University of New York, University at Buffalo, Buffalo, NY 14214-3079, USA; yc73@buffalo.edu; 4School of Medicine, Chang Gung University, Taoyuan 33302, Taiwan; 5Department of Child and Adolescent Psychiatry, Chang Gung Memorial Hospital, Kaohsiung Medical Center, Kaohsiung 83301, Taiwan

**Keywords:** COVID-19, sexual orientation, intention, vaccination, risk perception

## Abstract

The present study compared the levels of explicit and intrinsic intention to receive COVID-19 vaccination among sexual minority and heterosexual individuals and examined the association of explicit and intrinsic intentions with sexual orientation. We enrolled 171 sexual minority and 876 heterosexual individuals through a Facebook advertisement. The participants’ explicit and intrinsic intentions to receive COVID-19 vaccination and their risk perception of COVID-19 were measured. We discovered that sexual minority individuals had higher levels of explicit and intrinsic intention to receive vaccination relative to heterosexual individuals. Intrinsic intention was positively associated with explicit intention after the effects of demographic characteristics and risk perception of COVID-19 were controlled for. Sexual orientation did not moderate the association between explicit and intrinsic intentions. The present study determined the relationship between sexual orientation and intention to receive COVID-19 vaccination.

## 1. Introduction

### 1.1. Health of Sexual Minority Individuals during the Coronavirus Disease 2019 Pandemic

The coronavirus disease 2019 (COVID-19) pandemic has greatly affected people’s lives worldwide [1,2,3,4,5,6,7,8]. Having access to abundant economic, health care, and social support resources can help an individual to cope with the impact of the COVID-19 pandemic [9,10,11]. Research has revealed that minority individuals who are victimized by poverty, racism, and discrimination experience disproportionately adverse effects of the COVID-19 pandemic relative to nonminority individuals [12,13,14,15,16]; sexual minority individuals are no exception [17,18,19,20]. Compared with nonminority individuals, sexual minority individuals encounter more physiological risks (e.g., human immunodeficiency virus (HIV) and other sexually transmitted diseases; diabetes, hypertension, asthma, and substance abuse), mental health problems (e.g., depression and anxiety), financial and economic crisis (e.g., unemployment and salary cuts and sexual stigma and inequality (e.g., hate speech)), and poor social support (e.g., reduced support due to lockdown and social distancing) when they face the challenges of the COVID-19 pandemic [17,21,22,23]. Thus, for more effective intervention, it is necessary to further examine the health needs of sexual minority individuals during the COVID-19 pandemic.

### 1.2. Vaccination against COVID-19 in Sexual Minority Individuals

Equitable access to safe and effective vaccines is critical to ending the COVID-19 pandemic [24]. At present, government-led COVID-19 vaccination programs are underway [25,26], and seven COVID-19 vaccines have been approved by the World Health Organization for emergency use [27]. The experience of Israel indicated that vaccination against COVID-19 is effective in preventing symptomatic and asymptomatic COVID-19 infections and COVID-19-related hospitalizations, severe disease, and death [28]. Contrarily, vaccination refusal may result in seriously bad consequences. For example, rumors and conspiracy theories contributed to parental refusal or delay of childhood vaccines and then resulted in the resurgence of respiratory infectious diseases, such as measles in the US [29] and Europe [30].

Although these vaccines have been demonstrated to be safe and effective [31,32,33,34,35,36], COVID-19 vaccine hesitancy remains prevalent worldwide [37,38,39,40]. Examining the intention to get vaccinated against COVID-19 and the factors related to the intention is necessary. Research has demonstrated that sexual stigma and discrimination result in medical mistrust among sexual minority individuals [41,42]. Moreover, medical mistrust was significantly associated with decreased COVID-19 vaccine acceptance in sexual and gender minority individuals [43]. Contrarily, the longstanding problem of HIV infections within sexual minority communities may affect their attitudes toward COVID-19 vaccination. Using the Health Belief Model, a study proposed that health information on HIV prevention and treatment strategies can be applied to the COVID-19 pandemic [23]. No study has compared the level of the intention to get vaccinated against COVID-19 between sexual minority and heterosexual individuals.

### 1.3. Explicit and Intrinsic Intention to Receive COVID-19 Vaccination

A common method for assessing people’s intention to receive vaccination is the use of a single-item Likert scale or visual analog scale (VAS) [40,44]. The VAS has the advantage of being easy to administer and score. Risk perception assessments based on the Health Belief Model [45] are commonly used as a cognitive measure of an individual’s intention to receive COVID-19 vaccination [46,47]. However, the VAS measures only explicit (i.e., conscious) but not implicit (i.e., unconscious) intention toward vaccination; both types of intention influence vaccination intention.

Vallée-Tourangeau et al. applied the cognitive model of empowerment (CME) as an alternative method to examine people’s intrinsic intention to receive vaccination against respiratory infectious diseases (RIDs) [48]. The CME was originally used to assess the intrinsic task motivation of workers [49]. Vallée-Tourangeau et al. applied concepts in the CME to develop the Motors of Influenza Vaccination Acceptance (MoVac-Flu) scale, which measures an individual’s intrinsic intention to receive RID vaccination [48]. The MoVac-Flu scale comprises the following four cognitive assessments: (a) how important vaccination is to an individual, (b) how impactful vaccination is to an individual, (c) how knowledgeable an individual feels about vaccination, and (d) how autonomous an individual feels about their decision to receive vaccination [48]. The vaccination-oriented CME provides a conceptual framework for developing intervention programs to enhance people’s intrinsic motivation to receive COVID-19 vaccination [48].

Determining the factors related to explicit intention to get vaccinated against COVID-19 is essential to the development of intervention programs. In prevention motivation theory [50,51], the intrinsic intention to receive vaccination is treated as a construct of coping appraisal for COVID-19; this may entail a significant association between explicit and intrinsic intention to receive COVID-19 vaccination. In addition, risk perception, which is a construct of threat appraisal [50,51], may promote hygiene and social distancing behaviors against RIDs [52]. Research has revealed that sexual minority individuals have a lower risk perception than heterosexual individuals [53]. Further research is warranted to determine whether sexual minority and heterosexual individuals differ with respect to the associations of intrinsic intention to receive COVID-19 vaccination and risk perception with explicit intention.

### 1.4. Aims of the Present Study

The present study examined three research questions. First, are the levels of explicit and intrinsic intention to receive COVID-19 vaccination different between sexual minority and heterosexual individuals? Second, is intrinsic intention significantly associated with explicit intention when risk perception is accounted for? Third, are the associations between explicit and intrinsic intention to receive COVID-19 vaccination different between sexual minority and heterosexual individuals? Accordingly, this study hypothesized the following:

**Hypothesis** **1** **(H_1_).***Sexual minority individuals**have higher explicit and intrinsic intention to receive COVID-19 vaccination relative to heterosexual individuals*.

**Hypothesis** **2** **(H_2_).***Risk perception and intrinsic intention to receive COVID-19 vaccination is significantly associated with explicit intention*.

**Hypothesis** **3** **(H_3_).***The associations of risk perception and intrinsic intentions with explicit intention to receive COVID-19 vaccination differ between sexual minority and heterosexual individuals*.

## 2. Methods

### 2.1. Participants

This study enrolled participants per the procedure in a previous study [54]. Specifically, 1047 participants were enrolled through a Facebook advertisement that ran from 15 October 2020, to 21 December 2020. Prospective participants were included if they were aged ≥ 20 years and resided in Taiwan. As of 21 December 2020, 627 patients had COVID-19 in Taiwan and 7 have died from the disease [55]. No COVID-19 vaccines were available in Taiwan when the present study was being conducted. The Institutional Review Board of Kaohsiung Medical University Hospital approved this study (KMUHIRB-EXEMPT(I) 20200019).

### 2.2. Measures

#### 2.2.1. Explicit Intention to Receive COVID-19 Vaccination

Explicit intention to receive COVID-19 vaccination was assessed using a questionnaire item. The questionnaire item and corresponding scores are detailed in Appendix A. A higher score indicated a higher explicit intention to receive COVID-19 vaccination [45].

#### 2.2.2. Intrinsic Intention to Receive COVID-19 Vaccination

We used the Drivers of COVID-19 Vaccination Acceptance Scale (DrVac-COVID19S) to measure the participants’ self-reported intrinsic intention to receive COVID-19 vaccination [53]. The DrVac-COVID19S was adapted from the MoVac-Flu scale [48] and uses the same constructs that the MoVac-Flu scale uses to assess traits described in the CME [49]. The four cognitive traits measured by the DrVac-COVID19S are an individual’s values (three items that assess how much an individual cares about the purpose of COVID-19 vaccination uptake), perception of vaccination efficacy (three items that assess how much an individual believes in the effects of COVID-19 vaccination uptake in preventing COVID-19 infections), knowledge regarding vaccination (three items that assess how much knowledge an individual has regarding COVID-19 vaccination uptake), and autonomy (three items that assess how much confidence and control an individual has in accessing COVID-19 vaccination if they want to). The question items and corresponding scores are listed in Appendix A. A higher total score indicates a higher intrinsic intention to receive COVID-19 vaccination [56]. The Cronbach’s α for the scale in the present study was 0.867.

#### 2.2.3. Risk Perception

A five-item questionnaire was used to measure risk perception regarding contracting COVID-19 [57]. The five items assessed respondents’ concerns about developing flu-like symptoms, concerns about the possibility of contracting COVID-19, concerns about COVID-19, perception of likelihood of contracting COVID-19, and perception of likelihood of contracting COVID-19 compared with nonfamily members. The question items and corresponding scores are listed in Appendix A. A higher total score indicates higher risk perception. The Cronbach’s α scale in the present study was 0.704.

#### 2.2.4. Demographic Characteristics

Data on participants’ sexual orientation (whether an individual identified as heterosexual, homosexual, bisexual, pansexual, or asexual or were unsure about their sexual orientation), gender (female or male), age, and education level were collected. The participants were then sorted into sexual minority and heterosexual groups.

### 2.3. Data Analysis

Data analysis was performed using SPSS 24.0 (SPSS, Chicago, IL, USA). The sexual minority and heterosexual participants were compared with respect to gender, age, and education level using χ^2^ and *t* tests. The two groups’ explicit and intrinsic intention to receive COVID-19 vaccination and risk perception of COVID-19 were compared using a multivariate analysis of covariance (MANCOVA) in which gender, age, and education level were controlled for.

The association between explicit and intrinsic intention to receive COVID-19 vaccination among sexual minority and heterosexual individuals was examined using multiple regression in which risk perception was controlled for. To examine the moderating effect of sexual orientation, we analyzed the interaction between sexual orientation and intrinsic intention to receive COVID-19 vaccination using multiple regression to determine the association of this interaction term with explicit intention to receive COVID-19 vaccination. A two-tailed *p* value < 0.05 indicated statistical significance.

## 3. Results

The data on demographic characteristics, explicit and intrinsic intention to receive COVID-19 vaccination, and risk perception are presented in Table 1. In total, 171 sexual minority and 876 heterosexual individuals participated in this study. The results for the comparison of the sexual minority and heterosexual participants’ gender, age, education level, explicit and intrinsic intention to receive COVID-19 vaccination, and risk perception of COVID-19 are presented in Table 2. No differences in gender and educational level were observed between the two groups, but the heterosexual participants tended to be older. The MANCOVA results indicated a significant difference between heterosexual and sexual minority participants in their explicit and intrinsic intention and risk perception. The post hoc comparison indicated that the sexual minority participants had higher levels of explicit and intrinsic intention to receive COVID-19 vaccination compared with the heterosexual participants. These results supported H_1_. However, no difference in risk perception was observed between the two groups.

Table 3 presents the results of the multiple regression analysis examining the association between explicit and intrinsic intention to receive COVID-19 vaccination. The results of Model I indicated that, after controlling for the effects of demographic characteristics, risk perception was positively associated with explicit intention to receive COVID-19 vaccination. The Model II results indicated that intrinsic intention was positively associated with explicit intention. The value of adjusted R^2^ increased from 0.039 in Model I to 0.520 in Model II, indicating that intrinsic intention had a significantly stronger association with explicit intention compared with the variables in Model I. These results supported H_2_.

We further analyzed the interactions between sexual orientation, risk perception, and intrinsic intention in a multiple regression to examine the association of this interaction term with explicit intention (Model III). The results revealed that the interactions between sexual orientation and risk perception and between sexual orientation and intrinsic intention were not significantly associated with explicit intention, indicating that sexual orientation did not moderate the association of risk perception and intrinsic intentions with explicit intention to receive COVID-19 vaccination. These results did not support H_3_.

## 4. Discussion

The present study found that sexual minority individuals had higher levels of explicit and intrinsic intention to receive COVID-19 vaccination relative to heterosexual individuals. Intrinsic intention was positively associated with explicit intention after we controlled for the effects of demographic characteristics and risk perception of COVID-19. Sexual orientation did not moderate the association between explicit and intrinsic intention to receive COVID-19 vaccination.

### 4.1. Intention among Sexual Minority Individuals to Receive COVID-19 Vaccination

No differences in the level of risk perception of COVID-19 were observed between sexual minority and heterosexual individuals; however, sexual minority individuals had higher levels of explicit and intrinsic intention to receive COVID-19 vaccination relative to heterosexual individuals. Research reported that compared with heterosexual individuals, sexual minority individuals have greater medical needs and thus interact more with medical care providers [58]; sexual minority individuals may have greater access to information about preventive medicines, such as vaccines. Moreover, sexual minority individuals face the longstanding threat of HIV and other infectious diseases; therefore, they may already be more knowledgeable about and accepting of self-protective procedures (such as preexposure prophylaxis for infectious diseases) even before the start of the COVID-19 pandemic [23]. Although this study revealed a high intention among sexual minority individuals to receive COVID-19 vaccination, further research is warranted to determine whether the disadvantages resulting from financial status and social discrimination limit sexual minority individuals’ access to COVID-19 vaccination.

### 4.2. Factors Related to Explicit Intention to Receive COVID-19 Vaccination

The present study revealed that intrinsic intention to receive COVID-19 vaccination and risk perception of COVID-19 were positively associated with explicit intention after the effects of demographic characteristics were controlled for and that the associations did not significantly differ between sexual minority and heterosexual individuals. Intrinsic intention accounted for almost half of the variance of explicit intention, indicating that the components of intrinsic intention (including perceived importance and effects of vaccination, knowledge about vaccination, and autonomy in making decisions to receive vaccination) influenced individuals’ explicit intention to receive COVID-19 vaccination. These components of intrinsic intention ought to be targeted by intervention programs to improve the public vaccination rate.

Risk perception is the personal beliefs about the likelihood of suffering a disease [13]. Individuals who perceive a high risk of contracting a particular disease will adopt necessary measures to reduce the risk of developing it [15], whereas individuals with low perceived susceptibility may deny that they are at risk and be unlikely to engage in protective behaviors [15]. Therefore, enhancing people’s risk perception of COVID-19 is an important step to increase the vaccination rate during the COVID-19 pandemic. However, high perceived risk significantly affects the mental health of people during public health crises [50]. Governments and health professionals should actively promote awareness among the public regarding the threat of COVID-19 without evoking excessive worry.

However, intrinsic intention, risk perception, and demographic characteristics accounted for only 52% of the variance for explicit intention, indicating the presence of other factors that might have contributed to participants’ explicit intention to receive COVID-19 vaccination but were not examined in the present study; these other factors should be further investigated.

### 4.3. Strengths and Limitations

The present study is the first to compare sexual minority and heterosexual individuals’ levels of explicit and intrinsic intention to receive COVID-19 vaccination. The results aid the development of group-specific programs for enhancing individuals’ intention to receive COVID-19 vaccination. This study has several limitations. First, the participants were enrolled through a Facebook advertisement. Although enrolling participants using Facebook allows the researchers to quickly and easily reach out to a high number of prospective participants [59] during the pandemic, this enrollment method may limit how representative the participants are of the population [60]. Second, the number of sexual minority participants in the present study was small, which limited the possibility of segmenting the participants into sexual minority subgroups (e.g., lesbian, gay, or bisexual) for further analysis. Third, the intention to receive COVID-19 vaccination can be easily influenced by the severity of the COVID-19 pandemic. Longitudinal studies are required to examine changes in intention and behaviors relating to receiving vaccination. Fourth, although research reported sexual minority individuals have greater medical needs and interact more with medical care providers compared with heterosexual individuals [58] and thus we surmised that sexual minority individuals might have increased accesses to information about vaccines, we did not collect participants’ medical and psychopathological background.

## 5. Conclusions

The present study found that sexual minority individuals had higher levels of explicit and intrinsic intentions to receive COVID-19 vaccination compared with heterosexual individuals. The development of sexual orientation-specific programs is beneficial for enhancing individuals’ intention to receive COVID-19 vaccination. Intrinsic intention was revealed to be positively associated with explicit intention among both sexual minority and heterosexual individuals. The components of intrinsic intention ought to be targeted by intervention programs to increase the public’s intention to receive COVID-19 vaccination.

## Figures and Tables

**Table 1 ijerph-18-07260-t001:** Demographic characteristics, explicit and intrinsic intention to receive COVID-19 vaccination, and risk perception (*n* = 1047).

	*n* (%)	Mean (*SD*)	Range
Sexual orientation			
Heterosexual	876 (83.7)		
Sexual minority	171 (16.3)		
Gender			
Female	617 (58.9)		
Male	430 (41.1)		
Age (years)		35.5 (9.6)	21–70
Education level		3.2 (0.7)	0–5
Intention to get vaccinated against COVID-19			
Explicit intention		6.5 (2.6)	1–10
Intrinsic intention		61.9 (11.3)	24–90
Risk perception		17.6 (5.4)	5–38

SD: standard deviation.

**Table 2 ijerph-18-07260-t002:** Comparisons of heterosexual and sexual minority participants’ demographic characteristics, explicit and intrinsic intention to receive COVID-19 vaccination, and risk perception.

	Heterosexual (*n* = 876)	Sexual Minority (*n* = 171)	χ^2^ or *t* or *F* ^b^	*p*
Gender, *n* (%)				
Female	522 (59.6)	95 (55.6)	0.962	0.327
Male	354 (40.4)	76 (44.4)		
Age, mean (SD)	35.8 (9.4)	34.0 (10.5)	2.234	0.026
Education level, mean (SD)	3.2 (0.7)	3.2 (0.7)	0.611	0.541
Intention to get vaccinated against COVID-19, mean (SD) ^a^				
Explicit intention	6.5 (0.1)	7.3 (0.2)	14.371	<0.001
Intrinsic intention	61.9 (0.4)	64.0 (0.8)	4.981	0.026
Risk perception, mean (SD) ^a^	17.5 (0.2)	17.8 (0.4)	0.412	0.521

^a^: estimated marginal mean after controlling for sex, age, and educational level by conducting multivariate analysis of covariance; ^b^: multivariate analysis of covariance. SD: standard deviation.

**Table 3 ijerph-18-07260-t003:** Multiple regression analysis of factors related to intention to receive COVID-19 vaccination.

	Explicit Intention to Get Vaccinated								
	Model I			Model II			Model III		
	*B*	*SE*	*p*	*B*	*SE*	*p*	*B*	*SE*	*p*
Sexual minority ^a^	0.801	0.214	<0.001	0.458	0.152	0.003	1.646	1.023	0.108
Male ^b^	0.690	0.161	<0.001	0.006	0.115	0.955	0.010	0.115	0.932
Age	−0.007	0.008	0.404	0.003	0.006	0.639	0.002	0.006	0.683
Education level	0.185	0.115	0.108	0.028	0.081	0.732	0.028	0.081	0.728
Risk perception	0.049	0.015	0.001	0.065	0.010	<0.001	0.065	0.011	<0.001
Intrinsic intention to get vaccinated				0.164	0.005	<0.001	0.166	0.005	<0.001
Nonheterosexuals x risk perception							−0.019	0.014	0.177
Nonheterosexuals x intrinsic intention to get vaccinated							0.000	0.027	0.985
Adjusted *R*^2^	0.039			0.520			0.521		

^a^: heterosexuals as reference; ^b^: female as reference. SE: standard deviation.

## Data Availability

The data will be available upon reasonable request to the corresponding authors.

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
