# Peer review of "Explicit and Intrinsic Intention to Receive COVID-19 Vaccination among Heterosexuals and Sexual Minorities in Taiwan"

_ijerph, 2021, doi:10.3390/ijerph18147260_

Round 1

Reviewer 1 Report

Dear Authors,

Thank you for the opportunity to read and review your manuscript submitted to IJERPH. After reading the manuscript I am totally convinced that the research of the intention to receive COVID-19 vaccination among heterosexuals and sexual minorities is relevant and timely under the conditions of pandemic. The research has a firm theoretical and methodological base. The chosen statistical techniques are appropriate in order to reach the aim of the manuscript. Therefore I advise to accept the manuscript. 

Author Response

Dear Reviewer:

Thank you for your support!

Reviewer 2 Report

The introduction describes the disadvantage of minorities during the Covid19 pandemic. However, it calls for "examine the health needs of minorities during the pandemic". But your paper is about willingness to vaccination and not about health needs of minority population. Thus you need to explain the objective of the study. In addition, before you describe the intrinsic and extrensic motives for vaccination, you need to describe the vaccination problem of minority groups. What is the problem and why you want to study (the vaccination problem). 

The models of intention to adopt the vaccine are clear and well presented. But you need also to refer and what is new in your study, as you rely on known models. What is the added value of your study or how it is diferent from other studies in the field. This is an important question as vaccine hesitancy has a long long literature.

your basic argument is that minorities face more difficulties to cope with the pandemia, and therefore may face barriers to vaccination. Can you refer to studies that support this view?

Research questions and hypothesis are well presented and clear.

Measurment is clear and well presented as well as the analysis.

The discussion and conclusions are well presented. Yet I think that more discussion is needed on factors that might influence the willingness to get the vaccine and were not measured such as previous experience with vaccines, potential fear of side effects, etc.

overall an interesting paper.

Author Response

We appreciated your comments. As discussed below, we have revised our manuscript based on your suggestions. Please let us know if we need to provide anything else regarding this revision.

Comment 1

The introduction describes the disadvantage of minorities during the Covid19 pandemic. However, it calls for "examine the health needs of minorities during the pandemic". But your paper is about willingness to vaccination and not about health needs of minority population. Thus you need to explain the objective of the study. In addition, before you describe the intrinsic and extrinsic motives for vaccination, you need to describe the vaccination problem of minority groups. What is the problem and why you want to study (the vaccination problem).

Response

Thank you for your comment. In the revised manuscript we added the contents to explain why we considered the intention to get vaccination against COVID-19 in sexual minority individuals is essential to the health needs of minorities during the pandemic as below. Please refer to line 47-68.

“Equitable access to safe and effective vaccines is critical to ending the COVID-19 pandemic [24]. Government-led COVID-19 vaccination programs are underway [25,26], and seven COVID-19 vaccines have been approved by the World Health Organization for emergency use [27]. The experience of Israel indicated that vaccination against COVID-19 is effective in preventing symptomatic and asymptomatic COVID-19 infections and COVID-19-related hospitalizations, severe disease, and death [28]. Contrarily, vaccination refusal may result in seriously bad consequences. For example, rumors and conspiracy theories contributed to parental refusal or delay of childhood vaccines and then resulted in the resurgence of respiratory infectious diseases such as measle in the US [29] and Europe [30].

Although these vaccines have been demonstrated to be safe and effective [31-36], COVID-19 vaccine hesitancy remains prevalent worldwide [3740]. Examining the intention to get vaccination against COVID-19 and the factors related to the intention is necessary. Research has demonstrated that sexual stigma and discrimination result in medical mistrust among sexual minority individuals [41,42]. Moreover, medical mistrust was significantly associated with decreased COVID-19 vaccine acceptance in sexual and gender minority individuals [43]. Contrarily, the longstanding problem of HIV infections within sexual minority communities may affect their attitudes toward COVID-19 vaccination. Using the Health Belief Model, a study proposed that health information on HIV prevention and treatment strategies can be applied to the COVID-19 pandemic [23]. No study has compared the level of the intention to get vaccinated against COVID-19 between sexual minority and heterosexual individuals.

Comment 2

The models of intention to adopt the vaccine are clear and well presented. But you need also to refer and what is new in your study, as you rely on known models. What is the added value of your study or how it is different from other studies in the field. This is an important question as vaccine hesitancy has a long long literature.

Response

Thank you for your suggestion. Two issues regarding the intention to get vaccinated against COVID-19 in sexual minority individuals have not been well examined. The first is the comparison of intention between sexual minority and heterosexual individuals. The second is the factors related to the intention in sexual minority individuals. We added the introductions as below into the revised manuscript.

Although these vaccines have been demonstrated to be safe and effective [31-36], COVID-19 vaccine hesitancy remains prevalent worldwide [37–40]. Examining the intention to get vaccination against COVID-19 and the factors related to the intention is necessary. Research has demonstrated that sexual stigma and discrimination result in medical mistrust among sexual minority individuals [41,42]. Moreover, medical mistrust was significantly associated with decreased COVID-19 vaccine acceptance in sexual and gender minority individuals [43]. Contrarily, the longstanding problem of HIV infections within sexual minority communities may affect their attitudes toward COVID-19 vaccination. Using the Health Belief Model, a study proposed that health information on HIV prevention and treatment strategies can be applied to the COVID-19 pandemic [23]. No study has compared the level of the intention to get vaccinated against COVID-19 between sexual minority and heterosexual individuals.” Please refer to line 57-68.

“Determining the factors related to explicit intention to get vaccinated against COVID-19 is essential to the development of intervention programs. In prevention motivation theory [50,51], the intrinsic intention to receive vaccination is treated as a construct of coping appraisal for COVID-19; this may entail a significant association between explicit and intrinsic intention to receive COVID-19 vaccination. In addition, risk perception, which is a construct of threat appraisal [50,51], may promote hygiene and social distancing behaviors against RIDs [52]. Research has revealed that sexual minority individuals have a lower risk perception than heterosexual individuals [53]. Further research is warranted to determine whether sexual minority and heterosexual individuals differ with respect to the associations of intrinsic intention to receive COVID-19 vaccination and risk perception with explicit intention. Please refer to line 89-99.

Comment 3

Your basic argument is that minorities face more difficulties to cope with the pandemic, and therefore may face barriers to vaccination. Can you refer to studies that support this view?

Response

Thank you for your comment. In the revised manuscript we added the introduction for the difficulties that sexual minority individuals may encounter and what barriers to vaccination they may face as below.

Compared with nonminority individuals, sexual minority individuals encounter more physiological risks (e.g., human immunodeficiency virus [HIV] and other sexual transmitted diseases; diabetes, hypertension, asthma, and substance abuse), mental health problems (e.g., depression and anxiety), financial and economic crisis (e.g., unemployment and salary cuts, sexual stigma and inequality (e.g., hate speeches), and poor social support (e.g., reduced support due to lockdown and social distancing) when they face the challenges of the COVID-19 pandemic [17,21-23].” Please refer to 37-43.

“Research has demonstrated that sexual stigma and discrimination result in medical mistrust among sexual minority individuals [41,42]. Moreover, medical mistrust was significantly associated with decreased COVID-19 vaccine acceptance in sexual and gender minority individuals [43]. Contrarily, the longstanding problem of HIV infections within sexual minority communities may affect their attitudes toward COVID-19 vaccination. Using the Health Belief Model, a study proposed that health information on HIV prevention and treatment strategies can be applied to the COVID-19 pandemic [23]. No study has compared the level of the intention to get vaccinated against COVID-19 between sexual minority and heterosexual individuals. Please refer to 60-68.

Comment 4

The discussion and conclusions are well presented. Yet I think that more discussion is needed on factors that might influence the willingness to get the vaccine and were not measured such as previous experience with vaccines, potential fear of side effects, etc.

Response

Thank you for your comment. We rewrote the paragraph “4.2. Factors Related to Explicit Intention to Receive COVID-19 Vaccination” as below and focused the two factors related to explicit intention examined in the present study. Please refer to 237-260.

The present study revealed that intrinsic intention to receive COVID-19 vaccination and risk perception of COVID-19 were positively associated with explicit intention after the effects of demographic characteristics were controlled for and that the associations did not significantly differ between sexual minority and heterosexual individuals. Intrinsic intention accounted for almost half of the variance of explicit intention, indicating that the components of intrinsic intention (including perceived importance and effects of vaccination, knowledge about vaccination, and autonomy in making decisions to receive vaccination) influenced individuals’ explicit intention to receive COVID-19 vaccination. These components of intrinsic intention ought to be targeted by intervention programs to improve the public vaccination rate.

Risk perception is the personal beliefs about the likelihood of suffering a disease [13]. Individuals who perceive a high risk of contracting a particular disease will adopt necessary measures to reduce the risk of developing it [15], whereas individuals with low perceived susceptibility may deny that they are at risk and unlike to engage in protective behaviors [15]. Therefore, enhancing people’s risk perception of COVID-19 is an important step to increase the vaccination rate during the COVID-19 pandemic. However, high perceived risk significantly affects the mental health of people during public health crises [50]. Governments and health professionals should actively promote awareness among the public regarding the threat of COVID-19 without evoking excessive worry.

However, intrinsic intention, risk perception, and demographic characteristics accounted for only 52% of the variance for explicit intention, indicating the presence of other factors that might have contributed to participants’ explicit intention to receive COVID-19 vaccination but were not examined in the present study; these other factors should be further investigated.

Reviewer 3 Report

In the discussion, the authors state: "...we surmise that compared with heterosexual individuals, sexual minority individuals have greater medical needs and thus interact more with medical care providers."

If this statement is true, why did the authors not include information on the medical and psychopathological background of each patient in the research protocol?

Clarification is required!

Author Response

Comment

In the discussion, the authors state: "...we surmise that compared with heterosexual individuals, sexual minority individuals have greater medical needs and thus interact more with medical care providers." If this statement is true, why did the authors not include information on the medical and psychopathological background of each patient in the research protocol? Clarification is required!

Response

Thank you for your comment. We revised the statement as below. We did not collect participants’ medical and psychopathological background. Therefore, we listed it as one of limitations as below in this study.

  1. “Research reported that compared with heterosexual individuals, sexual minority individuals have greater medical needs and thus interact more with medical care providers [50]; sexual minority individuals may have greater access to information about preventive medicines such as vaccines. Please refer to line XXX.
  2. “Fourth, although research reported sexual minority individuals have greater medical needs and interact more with medical care providers compared with heterosexual individuals [50] and thus we surmised that sexual minority individuals might have the increased accesses to information about vaccines, we did not collect participants’ medical and psychopathological background. Please refer to line XXX.